# Saliency-Aware Model Merging

**Jungin Park**[1]   **Jiyoung Lee**[2] *   **Kwanghoon Sohn**[1] *

## Abstract

Model merging aims to consolidate multiple task-specific models fine-tuned on different datasets into a unified architecture that performs cross-domain proficiency. Current data-free model merging methods often struggle to scale as they rely on simple parameter-level heuristics that ignore inter-layer dependencies and non-uniform distribution of expertise. This work proposes **SA-Merging**, which is built upon connectivity-based saliency formulations from structural pruning (*e.g.*, SynFlow) and extends them to the data-free model merging setting. We define a saliency score over task vectors relative to a shared base model, and further introduce merge-aware modulation that incorporates agreement across experts to mitigate task interference. Based on this formulation, an iterative saliency-aware merging procedure progressively removes non-informative updates while preserving end-to-end connectivity. Furthermore, we extend **SA-Merging** to introduce rank-wise saliency decomposition for LoRA without compromising their structural integrity. Extensive experiments on vision and language tasks demonstrate the effectiveness of our saliency-based approach, further reducing the gap between data-free and test-time adaptation methods. Code will be publicly available.

## 1. Introduction

The pretraining-and-finetuning paradigm (Hu et al., 2022b; 2023) has catalyzed the proliferation of task- and domain-specialized "expert" models derived from a common foundation backbone such as LLaMA family (Touvron et al., 2023; Dubey et al., 2024; Adcock et al., 2026), Qwen family (Bai et al., 2023; Team et al., 2024; Yang et al., 2025) in NLP, and

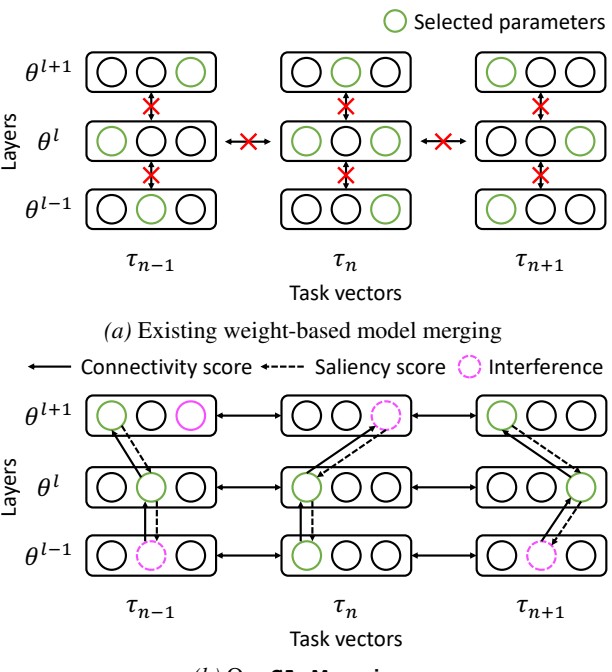

*(a)* Existing weight-based model merging

*(b)* Our **SA-Merging**

*Figure 1.* Comparison between (a) existing weight-based model merging methods (Ilharco et al., 2023; Matena & Raffel, 2022; Jin et al., 2022), which treat each parameter independently, and (b) our **SA-Merging** that takes the inter-layer interaction and the inter-model interference into account at once.

CLIP (Radford et al., 2021), ViT (Dosovitskiy et al., 2021) in computer vision. While this specialization achieves superior performance on isolated tasks, it introduces a significant infrastructure burden; storing and maintaining a growing library of distinct experts is computationally expensive and logistically cumbersome. More importantly, this compartmentalized approach creates *knowledge silos*, precluding the synergistic sharing of information across related domains.

Model merging (Ilharco et al., 2023; Matena & Raffel, 2022; Jin et al., 2022; Yadav et al., 2023) has emerged as a compelling paradigm for consolidating multiple fine-tuned experts directly into a unified parameter space to obtain a single multitask model, without the prohibitive cost of joint retraining. A most prevalent approach (Ilharco et al., 2023; Wortsman et al., 2022) represents each expert by a task vector, which is the parameter-wise displacement between

[1]Yonsei University, Seoul, South Korea [2]Ewha Womans University, Seoul, South Korea. Correspondence to: Kwanghoon Sohn <khsohn@yonsei.ac.kr>, Jiyoung Lee <lee.jiyoung@ewha.ac.kr>.

*Proceedings of the 43rd International Conference on Machine Learning*, Seoul, South Korea. PMLR 306, 2026. Copyright 2026 by the author(s).

a finetuned expert and the foundational base model, and then merges them with weighted averaging. Despite its efficiency, this linear weight-space interpolation is inherently limited. As the dimensionality and heterogeneity of tasks grow, simple element-wise accumulation of task vectors suffers from severe parameter interference. This interference arises because traditional methods treat experts' weights as i.i.d(independent and identically distributed) variables, failing to preserve the delicate functional alignment required for multitask proficiency. Consequently, the resulting merged models often exhibit a substantial performance gap compared to gold-standard multitask finetuning learning (MTL) (Yadav et al., 2023; Cheng et al., 2025; Sun et al., 2025).

To mitigate such interference, recent data-free approaches have introduced pruning and masking heuristics such as magnitude trimming and sign election (Yadav et al., 2023; Du et al., 2024; He et al., 2024; Sun et al., 2025; Yu et al., 2023). These methods, however, still predominantly operate under an element-wise independence assumption, treating each parameter as an isolated decision variable. This stands in contrast to the compositional hierarchy of deep neural networks, where functionality is not localized to individual weights but emerges through non-linear interactions across consecutive layers. By decoupling parameters from their structural context, existing heuristics often inadvertently prune weights that are critical for maintaining the coherent pathway of information flow across the network.

This paper investigates a fundamental yet overlooked question: *Can we identify which parameters to keep by accounting for inter-layer interactions within a data-free model merging framework?* In this work, we extend SynFlow (Tanaka et al., 2020), in which inter-layer interactions of parameters can be measured along a path from an input to an output node, to introduce an iterative saliency-aware model merging (**SA-Merging**) from multiple trained models. We leverage a connectivity score (Tanaka et al., 2020) as a structural proxy for task-wise end-to-end influence. The gradient of score derives a saliency score, *i.e.*, the importance of each parameter update. Intuitively, the gradient term quantifies the sensitivity of the network's end-to-end connectivity to changing a particular coordinate of an expert update, providing a data-free importance signal. To further mitigate task interference, we modulate this connectivity sensitivity with the current merged direction so that low-agreement coordinates receive low saliency even when their raw magnitudes are large. Building on this signal, **SA-Merging** recursively masks and removes noninformative updates, and then aggregates the remaining updates to construct a stable merged model.

We rigorously evaluate the efficacy of **SA-Merging** through a diverse suite of benchmarks, encompassing eight vision tasks with various vision transformer (ViT) backbones (Dosovitskiy et al., 2021), and eight natural language processing tasks with T5 (Raffel et al., 2020). Furthermore, we demonstrate the extensibility of our framework by generalizing connectivity-based saliency formulation to parameter-efficient tuning models (*e.g.*, LoRA (Hu et al., 2022b)). Experimental results show that the proposed saliency score successfully complements the existing magnitude-based merging basis. By effectively capturing the functional importance of parameter updates, **SA-Merging** consistently outperforms the state-of-the-art data-free methods in both vision and language tasks.

## 2. Related Work

Given a shared foundation model, training-free model merging has emerged as a powerful framework for constructing a unified checkpoint by directly orchestrating parameter weights. This approach circumvents the additional optimization while elegantly maintaining the inference cost of a single model. A foundational baseline in this domain is simple weight averaging (Wortsman et al., 2022), motivating work on when weight-space composition succeeds and how to make it reliable.

A widely used view in Ilharco et al. (2023)represents each expert as the base model plus a parameter update, enabling merging through simple parameter-space operations. Follow-up work improves this approach by learning non-uniform scaling of updates (Zhang et al., 2024) or selecting which parts of the update to keep using importance metrics (Bowen et al., 2024). ZipIt (Stoica et al., 2023) studies training-free merging across experts trained for different tasks, highlighting distinct cross-task behaviors. A central limitation is interference, typically arising from conflicting update directions and redundant or task-irrelevant changes. TIES-Merging addresses this by removing small-magnitude updates and resolving directional disagreements before merging (Yadav et al., 2023). Several recent methods further reduce conflicts via selective or localized rules, including CAT Merging (Sun et al., 2025), probabilistic masking (SeWA) (Wang et al., 2025a), and post-training layer scaling (LiNeS) (Wang et al., 2024). Sparsification provides another effective mechanism: DARE applies dropand-rescale to suppress destructive update components (Yu et al., 2023), while DELLA-Merging uses magnitude-based sampling to reduce conflicts (Deep et al., 2024). Other directions resolve interference in parameter-efficient merges via orthogonal subspaces (Zhang & Zhou, 2025), improve efficiency with frequency-domain transformations (Zheng & Wang, 2025), and tune merge coefficients through adaptive weighting or expert selection (Yang et al., 2023).

To scale to many experts, sparse or modular mechanisms restrict composition to subsets of parameters or modules

(Davari & Belilovsky, 2024; He et al., 2024; Lu et al., 2024). Because neural networks admit permutation symmetries, direct parameter-space merging can be sensitive to misalignment; re-basin methods mitigate this by merging modulo permutations (Ainsworth et al., 2022; Rinaldi et al., 2025). Finally, strictly data-free fusion considers the regime where only parameters are available (Jin et al., 2022), including task-vectorguided composition (Cheng et al., 2025), analyses of scaling behavior at larger expert counts (Yadav et al., 2024), theoretical failure modes as the number of experts grows (Wang et al., 2025b), and automated search over merging recipes (Akiba et al., 2025).

In contrast to existing works, we derive a saliency basis that couples consecutive layers and iteratively prunes low-saliency updates, yielding a strictly data-free merging procedure that jointly accounts for update magnitude, cross-layer connectivity, and directional agreement.

## 3. Method

### 3.1. Problem formulation

As shown in Figure 2, we consider a pretrained base model with parameters $\theta_0$ and $N$ independently fine-tuned experts $\{\theta_n\}_{n=1}^N$ for different tasks or domains. Following Ilharco et al. (2023), we represent each expert by a task vector $\tau_n$ defined by

$$\tau_n := \theta_n - \theta_0. \tag{1}$$

Data-free model merging aims to construct a single merged model $\theta^\star$ that performs well across tasks *without* access to any task-related data or test-time calibration samples. Many existing methods can be written as producing an aggregated expert via a (possibly parameter-wise) selection and reweighting rule:

$$\theta^\star = \theta_0 + \sum_{n=1}^N w_n \hat{\tau}_n, \qquad \text{where } \hat{\tau}_n = m_n \odot \tau_n. \tag{2}$$

$w_n$ is an expert weight and $m_n$ is a parameter-wise binary mask. For example, task arithmetic (Ilharco et al., 2023) corresponds to $m_n = \mathbb{1}$ with uniform $w_n$ applied to task vectors, while interference-aware methods (Yadav et al., 2023; Sun et al., 2025) can be interpreted as estimating a structured mask $m$ that suppresses redundant or conflicting parameters.

### 3.2. Saliency estimation

The main limitation of existing magnitude-based parameter selection rules lies in their localist bias. These methods implicitly assume that the importance of a parameter update is intrinsic to its own value. However, in deep models, the functional contribution of any single weight is fundamentally context-dependent, *i.e.*, its participation in the end-to-end

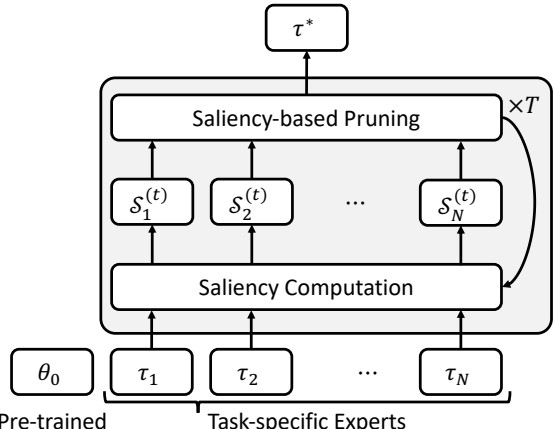

*Figure 2.* Overall procedure of **SA-Merging**. The framework begins with a shared base model $\theta_0$ and a set of task-specific experts (namely, task vector $\tau$). In each iteration $t$, saliency scores $\mathcal{S}_n^{(t)}$ are computed for each expert update, capturing their functional contribution to the global model connectivity.

pathways determined through *chains* of consecutive layers. For example, a large-magnitude update that is "blocked" by small weights in adjacent layers may have less functional saliency than a smaller update that resides on a high-capacity path. Motivated by SynFlow's connectivity score, where saliency is computed for a single model, we reformulate the connectivity-based saliency score with respect to task vectors and further modulate it using the aggregated merge direction to account for cross-expert agreement.

Let a model consist of $L$ consecutive parameter blocks, and let $\theta^l$ denote the parameters in $l$-th block (for simplicity, think of a representative weight matrix per block). We define a connectivity score $\mathcal{R}_n$, which acts a a data-free proxy for the total signal transmission capacity of the $n$-th task vector:

$$\mathcal{R}_n(\theta_0, \tau_n) := \mathbb{1}^\top \left( \prod_{l=1}^L \left| \theta_0^l + \tau_n^l \right| \right) \mathbb{1}, \tag{3}$$

where $\mathbb{1}$ is the all-ones vector and $|\cdot|$ represents the element-wise absolute value operator. Conceptually, $\mathcal{R}_n$ sums end-to-end path strengths across the network hierarchy. By absolute magnitudes, we ensure the score captures the potential magnitude of information flow, regardless of sign, so parameters that lie on many strong paths have larger influence on $\mathcal{R}_n$. In practice, we instantiate blocks at the granularity of consecutive layers (*e.g.*, transformer blocks or MLP layers) and apply $\mathcal{R}_n$ to the dominant weight tensors within each block (*e.g.*, projection matrices), optionally excluding biases and normalization parameters. This follows the intent of capturing cross-layer connectivity while keeping the functionality inexpensive to differentiate.

With the connectivity score, we estimate a saliency score $\mathcal{S}_n$

for each parameter in the $n$-th task vector. This is achieved by differentiating $\mathcal{R}_n$ *w.r.t.* its task vector and modulating the resulting gradient with the *aggregate direction*:

$$\mathcal{S}_n := \frac{\partial \mathcal{R}_n}{\partial \tau_n} \odot \sum_{i=1}^{N} \tau_i. \tag{4}$$

This formulation offers a dual-advantage for model merging;

- Structural sensitivity: The gradient term $\partial \mathcal{R}_n / \partial \tau_n$ weights each parameter by its contribution to the model's total connectivity. Parameters that are critical to maintaining the functional backbone of the expert receive higher saliency.

- Interference mitigation: By modulating with the aggregate direction $\sum_{i=1} \tau_i$, we perform implicit sign election. If an expert's update at a specific coordinate contradicts the collective agreement of the ensemble, the resulting product will be small (or negative), effectively downweighting the update as unreliable noise.

Through this mechanism, $\mathcal{S}_n$ effectively filters out parameters that are either structurally redundant or task-conflicting.

### 3.3. Iterative saliency-aware model merging

Rather than performing one-shot merging, which may overlook the shifting structural dependencies of the network, we follow the iterative pruning scheme in Tanaka et al. (2020) to progressively refine the merged parameter mask. We construct a binary mask $m_n$ for each task vector that retains only the most informative parameters. At each iteration $t \in \{1, \ldots, T\}$, we estimate task-wise saliency scores $\{\mathcal{S}_n^{(t)}\}$ based on the current state of the task vectors. We then update the task vectors by keeping the top $(1-p)$ fraction of parameters according to their saliency, where the pruning rate $p$ determines the sparsity level at each step. This process yields the updated mask $m_n^{(t)}$:

$$\tau_n^{(t+1)} := m_n^{(t)} \odot \tau_n^{(t)}, \tag{5}$$

where $\tau_n^{(0)}$ represents the initial task vector. After $T$ iterations, the final merged parameters $\theta^\star$ are obtained by aggregating all task vectors to the base parameter $\theta_0$:

$$\theta^\star = \theta_0 + \sum_{n=1}^{N} \tau_n^{(T)}. \tag{6}$$

This iterative refinement ensures that the merged model is composed of experts that are not only individually sparse but also structurally aligned to minimize cross-task interference. We first set the number of iterations $T$ to 10 as a fixed practical default. As shown in Figure 3(b), we observe a consistent trend that performance improves as increases across

**Algorithm 1** Saliency-aware model merging

---
1: **Input:** base parameters $\theta_0$; task vectors $\{\tau_n\}_{n=1}^{N}$; iterations $T$; prune ratio $p$
2: **for** $t = 1$ to $T$ **do**
3:     $\tau_t^* \leftarrow \sum_{i=1}^{N} \tau_i$
4:     **for** $n = 1$ to $N$ **do**
5:         $\mathcal{R}_n \leftarrow \mathbb{1}^\top \left( \prod_{l=1}^{L} \left| \theta_0^l + \tau_n^l \right| \right) \mathbb{1}$
6:         $\mathcal{S}_n \leftarrow \frac{\partial \mathcal{R}_n}{\partial \tau_n} \odot \tau_t^*$
7:         $m_n \leftarrow \text{TopKMask}(\mathcal{S}_n, 1-p)$
8:     **end for**
9:     $\tau_n \leftarrow m_n \odot \tau_n$ {Update task vectors}
10: **end for**
11: $\tau^* \leftarrow \sum_{i=1}^{N} \tau_n$ {Final merged task vector}
12: **Return:** merged model $\theta^\star \leftarrow \theta_0 + \tau^*$

---

all tasks. Furthermore, we observe that this trend holds consistently across different pruning ratios. Following prior sparsification and localized-merging literature such as TIES-merging (Yadav et al., 2023) and Localize-and-stitch (He et al., 2024), we then set a target number of parameters to keep after iterations to 10% of the total number of each task vector's parameters. Therefore, the iteration pruning ratio is set to $p = 0.2$ to satisfy $(1-p)^T \approx 0.1$. The complete procedure for **SA-Merging** is formalized in Algorithm 1. $\text{TopKMask}(\cdot, 1-p)$ selects the top $(1-p)$ fraction *within each tensor* (per-matrix masking), which avoids degenerate behavior where small tensors are removed entirely by global ranking.

### 3.4. Extend to LoRA experts

Beyond full-parameter merging, we extend our **SA-Merging** to low-rank adaptation (LoRA) (Hu et al., 2022b), which has become the de facto standard for scalable model specialization.

**LoRA as a task vector.** In LoRA fine-tuning (Hu et al., 2022a), the pretrained weights are frozen and each adapted linear layer $l$ is modified by a low-rank update. Let $W_0^l \in \mathbb{R}^{d_{\text{out}} \times d_{\text{in}}}$ denote the base weight and let expert $n$ provide LoRA factors $A_n^l \in \mathbb{R}^{r \times d_{\text{in}}}$ and $B_n^l \in \mathbb{R}^{d_{\text{out}} \times r}$ (with rank $r$). The induced weight update is

$$\Delta W_n^l = s B_n^l A_n^l, \qquad s := \alpha/r, \tag{7}$$

and we treat $\tau_n^l := \Delta W_n^l$ as the task-vector component for layer $l$ (and $\tau_n = 0$ for all non-adapted parameters). This converts LoRA experts into the same merging form as (2) while keeping the merge-time protocol strictly data-free.

**Rank-preserving saliency.** Our base algorithm masks individual coordinates of $\tau_n$. For LoRA, masking arbitrary entries of $\Delta W_n^l$ generally destroys the low-rank structure and cannot be represented by the same adapter rank. To preserve the LoRA parameterization, we use a *rank-preserving*

*Table 1.* Multi-task performance when merging CLIP ViT-B/32 models on the 8-task vision suite. Test-time/data-assisted methods use unlabeled test inputs, calibration corpora, or labeled validation sets (AdaMerging/AdaMerging++ (Yang et al., 2023), Representation Surgery (Yang et al., 2024)). Data-free baselines include weight averaging (Wortsman et al., 2022), Fisher (Matena & Raffel, 2022), RegMean (Jin et al., 2022), task arithmetic (Ilharco et al., 2023), TIES (Yadav et al., 2023), PCB (Du et al., 2024), and WUDI (Cheng et al., 2025). The best score is bold and second score is underlined.

| Method | SUN397 | Cars | RESISC45 | EuroSAT | SVHN | GTSRB | MNIST | DTD | **Avg.** |
|---|---|---|---|---|---|---|---|---|---|
| *Non-merging* | | | | | | | | | |
| Pretrained | 62.3 | 59.7 | 60.7 | 45.5 | 31.4 | 32.6 | 48.5 | 43.8 | 48.0 |
| Individual | 79.2 | 77.7 | 96.1 | 99.7 | 97.5 | 98.7 | 99.7 | 79.4 | 90.8 |
| Traditional MTL | 73.9 | 74.4 | 93.9 | 98.2 | 95.8 | 98.9 | 99.5 | 77.9 | 88.9 |
| *Test-time / data-assisted* | | | | | | | | | |
| AdaMerging | 64.5 | 68.1 | 79.2 | 93.8 | 87.0 | 91.9 | 97.5 | 59.1 | 80.1 |
| AdaMerging++ | 66.6 | 68.3 | 82.2 | 94.2 | 89.6 | 89.0 | 98.3 | 60.6 | 81.1 |
| Rep. Surgery | 63.8 | 59.9 | 83.3 | **97.9** | 87.0 | 87.0 | 98.6 | 69.4 | 80.9 |
| *Data-free model merging* | | | | | | | | | |
| Weight Avg. | 65.3 | 63.4 | 71.4 | 71.7 | 64.2 | 52.8 | 87.5 | 50.1 | 65.8 |
| Fisher Merging | 68.6 | 69.2 | 70.7 | 66.4 | 72.9 | 51.1 | 87.9 | 59.9 | 68.3 |
| RegMean | 65.3 | 63.5 | 75.6 | 78.6 | 78.1 | 67.4 | 93.7 | 52.0 | 71.8 |
| Task Arithmetic | 55.2 | 54.9 | 66.7 | 78.9 | 80.2 | 69.7 | 97.3 | 50.4 | 69.1 |
| TIES-Merging | 59.8 | 58.6 | 70.7 | 79.7 | 86.2 | 72.1 | 98.3 | 54.2 | 72.4 |
| PCB Merging | 66.7 | 65.5 | 78.5 | 79.3 | 86.4 | 77.1 | 98.2 | 59.1 | 76.3 |
| WUDI-Merging | 71.1 | 71.0 | 85.7 | 95.6 | 94.2 | 94.7 | 99.5 | 69.7 | 85.2 |
| **SA-Merging** (Ours) | **72.0** | **71.8** | **86.5** | 96.0 | **95.0** | **95.0** | **99.6** | **71.0** | **85.9** |

masking rule that prunes rank-1 components rather than individual matrix entries. We decompose the LoRA update into a sum of its rank-1 constituent components:

$$\Delta W_n^l \;=\; s \sum_{k=1}^{r} b_{n,k}^l (a_{n,k}^l)^\top, \tag{8}$$

where $b_{n,k}^l$ is the $k$-th column of $B_n^l$ and $a_{n,k}^l$ is the $k$-th row of $A_n^l$. Let $G_n^l := \frac{\partial \mathcal{R}(\theta_0 + \tau_n)}{\partial \Delta W_n^l}$ denote the connectivity sensitivity for layer $l$ (the corresponding block of $g_n$), and let $\overline{\Delta W}^l := \sum_i \Delta W_i^l$ be the aggregate merge direction for that layer.

**Computing sensitivities without materializing $\Delta W_n^l$.** Since $\Delta W_n^l = s B_n^l A_n^l$, the chain rule gives

$$\frac{\partial \mathcal{R}}{\partial B_n^l} \;=\; s\, G_n^l (A_n^l)^\top, \qquad \frac{\partial \mathcal{R}}{\partial A_n^l} \;=\; s\, (B_n^l)^\top G_n^l, \tag{9}$$

thereby one can obtain the needed quantities via automatic differentiation on the low-rank factors. The first-order effect of scaling component $k$ is the scalar

$$\gamma_{n,k}^l \;:=\; \langle G_n^l, b_{n,k}^l (a_{n,k}^l)^\top \rangle \;=\; (b_{n,k}^l)^\top G_n^l a_{n,k}^l, \tag{10}$$

and the agreement of that component with the aggregate direction is

$$\eta_{n,k}^l \;:=\; \left\langle \overline{\Delta W}^l, b_{n,k}^l (a_{n,k}^l)^\top \right\rangle \;=\; (b_{n,k}^l)^\top \overline{\Delta W}^l a_{n,k}^l. \tag{11}$$

We define a rank-wise merge-aware saliency as

$$s_{n,k}^l \;:=\; \left| \gamma_{n,k}^l \, \eta_{n,k}^l \right|. \tag{12}$$

This is the direct analogue of (4): $\gamma$ measures the connectivity importance along a rank-1 direction, and $\eta$ measures cross-expert agreement along the same direction. In practice, $\{\gamma_{n,k}^l\}_{k=1}^r$ can be computed efficiently as the diagonal of $(B_n^l)^\top G_n^l (A_n^l)^\top$ (and similarly for $\eta$).

**Rank-component masking update.** For each layer $l$ and expert $n$, we keep the top $(1-p)$ fraction of rank components by $s_{n,k}^l$ and define a binary mask vector $m_n^l \in \{0,1\}^r$. We then update the LoRA factors by zeroing the pruned components:

$$B_n^l \leftarrow B_n^l \operatorname{Diag}(m_n^l), \qquad A_n^l \leftarrow \operatorname{Diag}(m_n^l)\, A_n^l. \tag{13}$$

This preserves the LoRA form with rank at most $r$ while implementing the same iterative pruning structure as shown in Algorithm 2. Finally, we sum the remaining LoRA updates across experts to obtain the merged adapter (or equivalently a merged weight update) for each adapted layer.

**Optional post-hoc compression.** If one instead materializes the merged matrix update $\Delta W^{l,\star}$ (e.g., by applying element-wise masks on $\Delta W_n^l$), a low-rank adapter can be recovered without data via truncated SVD: $\Delta W^{l,\star} \approx s\, B^{l,\star} A^{l,\star}$ with rank $r'$ chosen for a desired parameter budget. This step is purely algebraic and does not use any task data.

## 4. Experiments

We evaluate **SA-Merging** under a strict *data-free* protocol: the merging process operates exclusively on the pretrained

*Table 2.* Multi-task performance when merging CLIP ViT-L/14 models on the 8-task vision suite. Test-time/data-assisted methods use unlabeled test inputs, calibration corpora, or labeled validation sets (AdaMerging/AdaMerging++ (Yang et al., 2023), Representation Surgery (Yang et al., 2024)). Data-free baselines include weight averaging (Wortsman et al., 2022), Fisher (Matena & Raffel, 2022), RegMean (Jin et al., 2022), task arithmetic (Ilharco et al., 2023), TIES (Yadav et al., 2023), PCB (Du et al., 2024), and WUDI (Cheng et al., 2025).

| Method | SUN397 | Cars | RESISC45 | EuroSAT | SVHN | GTSRB | MNIST | DTD | **Avg.** |
|---|---|---|---|---|---|---|---|---|---|
| *Non-merging* | | | | | | | | | |
| Pretrained | 66.8 | 77.7 | 71.0 | 59.9 | 58.4 | 50.5 | 76.3 | 55.3 | 64.5 |
| Individual | 82.3 | 92.4 | 97.4 | 100.0 | 98.1 | 99.2 | 99.7 | 84.1 | 94.2 |
| Traditional MTL | 80.8 | 90.6 | 96.3 | 96.3 | 97.6 | 99.1 | 99.6 | 84.4 | 93.5 |
| *Test-time / data-assisted* | | | | | | | | | |
| AdaMerging | 79.0 | 90.3 | 90.8 | 96.2 | 93.4 | 98.0 | 99.0 | 79.9 | 90.8 |
| AdaMerging++ | 79.4 | 90.3 | 91.6 | 97.4 | 93.4 | 97.5 | 99.0 | 79.2 | 91.0 |
| Rep. Surgery | 75.7 | 84.4 | 93.1 | 98.8 | 91.3 | 93.4 | 99.1 | 76.1 | 89.0 |
| *Data-free model merging* | | | | | | | | | |
| Weight Avg. | 72.1 | 81.6 | 82.6 | 91.9 | 78.2 | 70.7 | 97.1 | 62.8 | 79.6 |
| Fisher Merging | 69.2 | 88.6 | 87.5 | 93.5 | 80.6 | 74.8 | 93.3 | 70.0 | 82.2 |
| RegMean | 73.3 | 81.8 | 86.1 | 97.0 | 88.0 | 84.2 | 98.5 | 60.8 | 83.7 |
| Task Arithmetic | 73.9 | 82.1 | 86.6 | 94.1 | 87.9 | 86.7 | 98.9 | 65.6 | 84.5 |
| TIES-Merging | 76.5 | 85.0 | 89.3 | 95.7 | 90.3 | 83.3 | 99.0 | 68.8 | 86.0 |
| PCB Merging | 76.8 | 86.2 | 89.4 | 96.5 | 88.3 | 91.0 | 98.6 | 73.6 | 87.5 |
| WUDI-Merging | 81.0 | 91.0 | 94.2 | 99.2 | 96.3 | 98.1 | 99.6 | 81.2 | 92.6 |
| **SA-Merging** (Ours) | **82.0** | **91.8** | **95.0** | **99.4** | **97.0** | **98.6** | **99.7** | **83.5** | **93.4** |

base parameters $\theta_0$ and the fine-tuned expert parameters $\{\theta_n\}$ (equivalently, task vectors $\{\tau_n\}$). Our approach bypasses the need for training, validation, or even unlabeled calibration data to determine hyperparameters or importance weights. We still follow standard evaluation protocols on the test sets of each task, while the *merging procedure is data-free*. By eliminating reliance on auxiliary data, `SA-Merging` ensures a highly practical and scalable merging procedure, independent of data availability or privacy constraints.

### 4.1. Benchmarks and protocol

**Base benchmark suite.** To match recent data-free merging evaluations (Cheng et al., 2025), we include *all* benchmark families used in their experimental setting: (i) an 8-task vision suite with CLIP ViT backbones (Radford et al., 2021); (ii) the 8-task GLUE benchmark for discriminative language understanding (Wang et al., 2018) with RoBERTa encoders (Liu et al., 2019); (iii) a decoder-based, instruction/math/code merging suite evaluated on AlpacaEval (Dubois et al., 2024), GSM8K (Cobbe et al., 2021), MATH (Hendrycks et al., 2021), HumanEval (Chen et al., 2021), and MBPP (Austin et al., 2021); and (iv) an additional LoRA-merging setting with Flan-T5-base (Chung et al., 2024) experts.

**Vision (CLIP ViT).** We start from pretrained CLIP image encoders and fine-tune $N=8$ task experts on SUN397 (Xiao et al., 2016; 2010), Stanford Cars (Krause et al., 2013), RESISC45 (Cheng et al., 2017), EuroSAT (Helber et al.,

2019), SVHN (Netzer et al., 2011), GTSRB (Stallkamp et al., 2012), MNIST (LeCun et al., 2002), and DTD (Cimpoi et al., 2014). We report top-1 accuracy per task and macro-average for ViT-B/32, ViT-B/16, and ViT-L/14.

**Language (GLUE).** We merge $N=8$ GLUE experts (CoLA, MNLI, MRPC, QNLI, QQP, RTE, SST-2, STS-B) for RoBERTa-Base and RoBERTa-Large (Liu et al., 2019), reporting the average normalized score following (Cheng et al., 2025) (100 corresponds to the corresponding single-task expert).

**LoRA merging.** To assess the broader applicability of our framework, we evaluate `SA-Merging` in the context of parameter-efficient fine-tuning (PEFT). Specifically, we consolidate eight distinct LoRA (Hu et al., 2022b) experts, each fine-tuned on a separate task from the GLUE benchmark suite using the Flan-T5-base (Chung et al., 2024) backbone. In addition, we employ Qwen-14B's (Bai et al., 2023) LoRA experts fine-tuned on four tasks, including MMLU (Hendrycks et al., 2020), TruthfulQA (Lin et al., 2021), BBQ (Parrish et al., 2021), and CNN/DailyMail (Hermann et al., 2015).

### 4.2. Baselines

We compare against training-free merging methods that operate on full parameters or task vectors: simple weight averaging (Wortsman et al., 2022), task arithmetic (Ilharco et al., 2023), TIES-Merging (Yadav et al., 2023), DARE (Yu et al., 2023), WUDI-Merging (Cheng et al., 2025), PCB-

*Table 3.* Multi-task performance when merging RoBERTa experts on the 8-task GLUE benchmark. We report the average normalized score following (Cheng et al., 2025). Data-free baselines include weight averaging (Wortsman et al., 2022), task arithmetic (Ilharco et al., 2023), TIES (Yadav et al., 2023), DARE (Yu et al., 2023), PCB (Du et al., 2024), and WUDI (Cheng et al., 2025). Each task score is reported in Table B2 and Table B3.

| Method | RoBERTa-Base | RoBERTa-Large |
|---|---|---|
| *Non-merging* | | |
| Pretrained | 41.7 | 38.2 |
| Individual | 100.0 | 100.0 |
| *Data-free model merging* | | |
| Weight Avg. | 52.6 | 53.3 |
| Task Arithmetic | 67.8 | 70.9 |
| TIES-Merging | 64.7 | 72.4 |
| TA + DARE | 63.7 | 70.9 |
| TIES + DARE | 65.6 | 72.8 |
| PCB Merging | 76.5 | 79.0 |
| WUDI-Merging | 85.3 | 88.8 |
| **SA-Merging** (Ours) | **87.1** | **90.2** |

Merging (Du et al., 2024), Localize-and-Stitch (He et al., 2024), and CAT Merging (Sun et al., 2025). For completeness, we also report test-time/data-assisted baselines that use unlabeled calibration inputs or labeled validation sets at merge time, including AdaMerging (Yang et al., 2023), representation surgery (Yang et al., 2024), activation-guided consensus merging (ACM) (Yao et al., 2025), and DF-Merge (Lee et al., 2025); these results are explicitly grouped and labeled as non-data-free in the tables.

## 4.3. Main results

### 4.3.1. VISION TASKS

Table 1 and Table 2 show the effectiveness of our **SA-Merging** on multiple vision tasks. On the ViT-B/32 backbone, **SA-Merging** demonstrates an average accuracy of 85.9%, outperforming the best data-free baseline, WUDI-Merging (85.2%), and significantly surpassing data-assisted methods like AdaMerging++ (81.1%). This trend continues with the larger ViT-L/14 model, where **SA-Merging** reaches an average accuracy of 93.4%, nearly matching the performance of traditional MTL (93.5%) and individual experts (94.2%), without requiring any joint training or data access. The results with ViT-B/16 backbone are reported in Table B1. Across the ViT series, **SA-Merging** consistently improves over weight averaging and task arithmetic, and matches or exceeds strong sparsification-based baselines.

### 4.3.2. LANGUAGE (GLUE)

In Table 3, **SA-Merging** achieves a new state-of-the-art for data-free model merging with average normal-

ized accuracy of 87.1% on RoBERTa-Base and 90.2% on RoBERTa-Large, surpassing the prior state-of-the-art WUDI-Merging with a 1.8p and 1.4p margins on RoBERTa-BASE and -Large, respectively. The performance gap between **SA-Merging** and standard heuristics is substantial, exceeding +20% over TIES-Merging (64.7%) and Task Arithmetic (67.8%). Furthermore, the advantage of structural saliency remains consistent as the model backbone scales, proving its effectiveness in managing the increased interference found in larger parameter spaces. These findings suggest that a robust structural saliency signal can effectively substitute for empirical data feedback, successfully closing the gap between data-free and data-dependent merging regimes.

### 4.3.3. LoRA MERGING

Merging low-rank adapters trained on LLM is particularly challenging due to the highly compressed nature of the updates, which amplifies the risk of functional collapse when experts are naively aggregated. Table 4 shows our connectivity-based saliency scores effectively identify the most critical low-rank directions across tasks, although improvements are smaller than full-parameter merging due to the low-rank parameterization and its initialization-induced noise. Nevertheless, these results underscore the robust extensibility of **SA-Merging**, establishing it as a versatile, data-free framework capable of preserving functional pathways even within highly constrained, low-rank parameter spaces.

The results in Table 5 show that **SA-Merging** consistently outperforms the baselines under a larger-scale PEFT setting. This indicates that the proposed rank-wise saliency formulation is not restricted to smaller models, such as Flan-T5-base, but also generalizes to larger language models.

Together with the strong performance on ViT full-parameter merging and language LoRA merging, these results further support the broader claim that **SA-Merging** is not modality-specific, but provides a general formulation for saliency-aware model merging.

## 4.4. Analysis

### 4.4.1. EFFECT OF PRUNING RATIO AND ITERATIONS

We sweep the pruning ratio $p$ and number of iterations $T$ and observe that moderate sparsification yields the best trade-off: overly aggressive pruning removes task-critical connectivity paths, while insufficient pruning leaves sign-conflicted coordinates intact. As shown in Figure 3(a), we observe that the merged model reaches its best performance at $p = 0.2$ by balancing knowledge preservation and interference mitigation. Figure 3(b) shows that **SA-Merging** exhibits a rapid and synchronized convergence across all tasks, reaching

*Table 4.* LoRA merging results on the 8-task GLUE suite using Flan-T5-base experts, following the benchmark family in (Cheng et al., 2025). Test-time/data-assisted methods use unlabeled test inputs, calibration corpora, or labeled validation sets (AdaMerging++ (Yang et al., 2023), ACM (Yao et al., 2025), DF-Merge (Lee et al., 2025)). Data-free baselines include task arithmetic (Ilharco et al., 2023), TIES (Yadav et al., 2023), PCB (Du et al., 2024), and WUDI (Cheng et al., 2025).

| Method | CoLA | MNLI | MRPC | QNLI | QQP | RTE | SST-2 | STS-B | **Avg.** |
|---|---|---|---|---|---|---|---|---|---|
| *Non-merging* | | | | | | | | | |
| Individual | 69.1 | 82.7 | 85.5 | 90.9 | 84.0 | 84.4 | 92.9 | 87.4 | 84.6 |
| *Test-time / data-assisted* | | | | | | | | | |
| AdaMerging++ | 69.1 | 60.3 | 78.4 | 90.0 | 83.6 | 79.1 | 91.6 | 74.1 | 78.3 |
| *Data-free* | | | | | | | | | |
| Weight Avg. | 67.0 | 55.0 | 77.0 | 89.0 | 83.0 | 77.0 | 91.0 | 70.0 | 76.1 |
| Task Arithmetic. | 67.5 | 55.5 | 78.0 | 89.2 | 83.2 | 78.0 | 91.2 | 71.0 | 76.7 |
| TIES | 68.3 | 56.3 | 79.4 | 89.8 | 83.7 | 79.4 | 91.6 | 71.2 | 77.5 |
| PCB | 68.0 | 60.0 | 80.0 | 90.0 | 83.7 | 79.0 | 91.8 | 79.5 | 79.0 |
| WUDI | 68.6 | 79.0 | 77.7 | 87.2 | 83.1 | 75.8 | **93.2** | 85.0 | 81.2 |
| **SA-Merging** (Ours) | **69.8** | **68.0** | **84.0** | **91.5** | **84.6** | **83.0** | **93.2** | **85.5** | **82.5** |

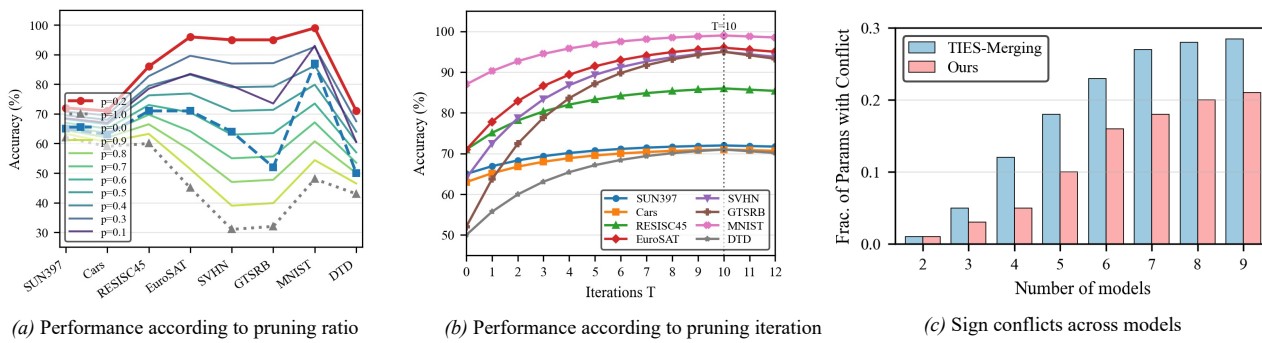

*(a) Performance according to pruning ratio*  *(b) Performance according to pruning iteration*  *(c) Sign conflicts across models*

*Figure 3.* Analysis on (a) pruning ratio, (b) pruning iteration, and (c) sign conflict across models.

*Table 5.* LoRA merging results on four tasks, including MMLU (Hendrycks et al., 2020), TruthfulQA (Lin et al., 2021), BBQ (Parrish et al., 2021), and CNN/DailyMail (Hermann et al., 2015), using Qwen-14B (Bai et al., 2023) experts, following the benchmark family in (Cheng et al., 2025).

| Method | MMLU | TruthfulQA | BBQ | CNN | Avg. |
|---|---|---|---|---|---|
| Individual | 68.35 | 53.34 | 93.53 | 19.46 | 58.67 |
| Task Arithmetic | 67.56 | 52.33 | 78.38 | **20.54** | 54.70 |
| TIES Merging | 69.38 | 52.03 | 81.06 | 15.91 | 54.62 |
| WUDI-Merging | 69.17 | **55.71** | 80.56 | 17.33 | 55.69 |
| **SA-Merging** (Ours) | **69.87** | 55.60 | **81.35** | 18.48 | **56.33** |

peak performance within approximately 10 iterations. This indicates that our saliency signal effectively identifies and mitigates task interference through iterative refinement.

#### 4.4.2. CONFLICT LOCALIZATION.

We further analyze where **SA-Merging** prunes parameters and find that pruned coordinates concentrate in layers with high sign disagreement across experts, while preserved coordinates align with strong connectivity gradients. This

complements prior sign-based conflict analyses (Yadav et al., 2023; Sun et al., 2025) by introducing an explicit connectivity weighting. As shown in Figure 3(c), our **SA-Merging** introduces significantly fewer parameter conflicts between models.

## 5. Conclusion

We studied a strictly data-free model merging, where a single multitask checkpoint must be composed from a shared base model and multiple fine-tuned experts without access to any training or validation data during merging. We utilized the connectivity-aware saliency basis of the task vector that couples consecutive layers and yields parameter rankings that reflect end-to-end influence, and iterative saliency pruning that constructs a refined task-wise parameter mask for composing task vectors. Our **SA-Merging** superiorly navigates both full-parameter and low-rank adaptation settings, consistently outperforming prior arts in vision, language, and Flan-T5-base LoRA experts consolidation.

**Limitations.** Our connectivity function is a structural surrogate defined on parameters; while it is data-free and dif-

ferentiable, it is not guaranteed to correlate perfectly with downstream accuracy for every architecture and task. The computational cost scales with the number of experts and the number of pruning iterations, and careful engineering is required for very large models. Finally, our current write-up focuses on strictly data-free merging; extending the approach to incorporate small amounts of calibration data (when available) is an important direction but departs from the strict setting.

## Impact Statement

This work introduces `SA-Merging`, a structural framework for efficient model merging that significantly advances the sustainability and accessibility of large-scale AI. By enabling model consolidation without the need for joint retraining, our method drastically reduces the energy consumption and carbon footprint associated with developing multitask systems. Furthermore, as a purely data-free approach, it facilitates the secure integration of expertise across models without compromising the privacy of sensitive training data, which is paramount in domains such as healthcare and finance.

## Acknowledgements

This work was supported by the National Research Foundation of Korea(NRF) grant funded by the Korea government(MSIT) (RS-2025-02216328).

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

# A. Additional Details

## A.1. Implementation details

The connectivity score (3) is defined over the dominant linear operators. For Transformer architectures, we apply it to the major projection matrices (attention and MLP). For parameters that do not participate in these products (*e.g.*, LayerNorm scale/shift and biases), we keep them unpruned by default.

## A.2. LoRA algorithm

We provide the overall merging process with LoRA in Algorithm 2.

---

**Algorithm 2** LoRA-based `SA-Merging`

---

1: **Input:** Base parameters $\theta_0$, LoRA adapters $\{A_n^{(0)}, B_n^{(0)}\}_{n=1}^N$, iterations $T$, pruning rate $p$.
2: **Initialize:** Rank-wise masks $m_n^{l,(1)} \leftarrow \mathbf{1}_r$ for all layers $l$ and tasks $n$.
3: **for** $t = 1$ to $T$ **do**
4:    $\overline{\Delta W}^{(t)} \leftarrow \sum_{i=1}^N s B_i^{(t-1)} A_i^{(t-1)}$ {Compute aggregate low-rank direction}
5:    **for** $n = 1$ to $N$ **do**
6:       Compute gradient: $G_n^{(t)} \leftarrow \frac{\partial \mathcal{R}_n}{\partial (\theta_0 + s B_n^{(t-1)} A_n^{(t-1)})}$
7:       $\gamma_{n,k} \leftarrow \mathrm{diag}((B_n^{(t-1)})^\top G_n^{(t)} (A_n^{(t-1)})^\top)$ {Rank-wise connectivity importance}
8:       $\eta_{n,k} \leftarrow \mathrm{diag}((B_n^{(t-1)})^\top \overline{\Delta W}^{(t)} (A_n^{(t-1)})^\top)$ {Rank-wise agreement}
9:       $s_{n,k} \leftarrow |\gamma_{n,k} \eta_{n,k}|$ {Compute rank-wise saliency}
10:      Update mask: $m_n^{(t)} \leftarrow \mathrm{TopKMask}(1 - p)$ components based on $s_{n,k}$
11:      Update factors: $B_n^{(t)} \leftarrow B_n^{(t-1)} \mathrm{Diag}(m_n^{(t)})$, $A_n^{(t)} \leftarrow \mathrm{Diag}(m_n^{(t)}) A_n^{(t-1)}$
12:    **end for**
13: **end for**
14: **Merge:** $\theta^\star \leftarrow \theta_0 + \lambda \sum_{n=1}^N s B_n^{(T)} A_n^{(T)}$
15: **Return:** $\theta^\star$

---

## A.3. Additional interpretation of connectivity gradients

For a sequence of linear layers with weights $\{\theta^l\}_{l=1}^L$, define the forward "flow" vectors

$$a^0 := \mathbf{1}, \qquad a^l := |\theta^l| \, a^{l-1}, \tag{14}$$

and the backward flow vectors

$$b^L := \mathbf{1}, \qquad b^{l-1} := |\theta^l|^\top b^l. \tag{15}$$

Then $\mathcal{R}(\theta) = \mathbf{1}^\top a^L = b^{l\top} a^l$, and the gradient satisfies a prefix–suffix factorization: entries of $\partial \mathcal{R}/\partial |\theta^l|$ are proportional to $b^l (a^{l-1})^\top$ (Tanaka et al., 2020). This makes the sensitivity large for parameters that connect large upstream and downstream flows, aligning saliency with inter-layer connectivity.

# B. Additional results

**Performance with CLIP ViT-B/16 on vision tasks.**

We provide the result of ViT-B/16 in Table B1. The evaluation on the 8-task vision suite using the CLIP ViT-B/16 backbone further validates the robustness and precision of `SA-Merging` in high-resolution regimes. As shown in Table B1, `SA-Merging` consistently outperforms strong data-free baselines, providing a massive performance leap of up to 16p compared to standard heuristics (Ilharco et al., 2023; Yadav et al., 2023).

**Per-task performance on the GLUE benchmark.**

Per-task normalized scores for RoBERTa-Base and RoBERTa-Large on the 8-task GLUE benchmark in Table B2 and Table B3.

*Table B1.* Multi-task performance when merging CLIP ViT-B/16 models on the 8-task vision suite. Test-time/data-assisted methods use unlabeled test inputs, calibration corpora, or labeled validation sets (AdaMerging/AdaMerging++ (Yang et al., 2023), Representation Surgery (Yang et al., 2024)). Data-free baselines include weight averaging (Wortsman et al., 2022), Fisher (Matena & Raffel, 2022), RegMean (Jin et al., 2022), task arithmetic (Ilharco et al., 2023), TIES (Yadav et al., 2023), PCB (Du et al., 2024), and WUDI (Cheng et al., 2025).

| Method | SUN397 | Cars | RESISC45 | EuroSAT | SVHN | GTSRB | MNIST | DTD | **Avg.** |
|---|---|---|---|---|---|---|---|---|---|
| *Non-merging* | | | | | | | | | |
| Pretrained | 63.8 | 64.6 | 65.7 | 54.5 | 52.0 | 43.3 | 51.7 | 45.1 | 55.0 |
| Individual | 81.8 | 86.8 | 96.9 | 99.7 | 97.8 | 99.1 | 99.7 | 82.0 | 92.9 |
| *Test-time / data-assisted* | | | | | | | | | |
| AdaMerging | 64.4 | 64.2 | 75.4 | 86.7 | 86.3 | 86.7 | 97.6 | 46.9 | 76.0 |
| AdaMerging++ | 70.2 | 80.7 | 81.6 | 94.8 | 91.6 | 95.8 | 98.5 | 66.2 | 84.9 |
| Rep. Surgery | 68.3 | 72.3 | 88.7 | 97.7 | 91.0 | 89.5 | 98.9 | 72.9 | 84.9 |
| DF-Merge | 76.0 | 83.0 | 91.2 | 98.1 | **95.8** | 96.8 | **99.5** | **76.0** | 89.6 |
| *Data-free model merging* | | | | | | | | | |
| Weight Avg. | 67.7 | 70.0 | 75.3 | 79.5 | 74.9 | 60.1 | 94.4 | 43.8 | 70.7 |
| Fisher Merging | 68.5 | 69.9 | 75.2 | 80.4 | 73.2 | 61.2 | 94.5 | 50.7 | 71.7 |
| RegMean | 69.1 | 71.6 | 77.6 | 88.8 | 83.7 | 70.2 | 96.9 | 54.6 | 76.6 |
| Task Arithmetic | 61.1 | 65.9 | 74.0 | 76.2 | 88.0 | 73.9 | 98.4 | 53.0 | 73.8 |
| TIES-Merging | 69.1 | 72.5 | 80.5 | 84.0 | 85.0 | 71.5 | 98.1 | 54.9 | 77.0 |
| PCB Merging | 70.5 | 73.0 | 82.0 | 86.0 | 89.0 | 80.0 | 98.6 | 65.0 | 80.5 |
| WUDI-Merging | 75.7 | 82.5 | 90.7 | 98.0 | 95.4 | 96.6 | 99.4 | 74.7 | 89.1 |
| **SA-Merging** (Ours) | **76.8** | **83.5** | **92.0** | **98.2** | 95.8 | **97.0** | 99.5 | 76.0 | **89.8** |

*Table B2.* Per-task multi-task performance when merging RoBERTa-Base experts on the 8-task GLUE benchmark.

| Method | CoLA | SST-2 | MRPC | STS-B | QQP | QNLI | MNLI | RTE | Avg. |
|---|---|---|---|---|---|---|---|---|---|
| *Non-merging* | | | | | | | | | |
| Pretrained | 0.0 | 53.8 | 85.0 | 4.0 | 37.5 | 53.1 | 37.1 | 71.2 | 41.7 |
| Individual | 100.0 | 100.0 | 100.0 | 100.0 | 100.0 | 100.0 | 100.0 | 100.0 | 100.0 |
| *Data-free model merging* | | | | | | | | | |
| Weight Averaging | 0.0 | 59.2 | 85.8 | 47.0 | 45.4 | 63.9 | 48.0 | 71.2 | 52.6 |
| Task Arithmetic | 8.4 | 88.3 | 89.6 | 32.8 | 82.0 | 85.4 | 75.5 | 80.4 | 67.8 |
| TIES-Merging | 31.8 | 88.9 | 86.2 | 10.9 | 61.1 | 85.9 | 83.0 | 69.6 | 64.7 |
| Task Arithmetic (w/ DARE) | 0.0 | 88.1 | 86.6 | 30.2 | 84.3 | 79.1 | 64.0 | 77.2 | 63.7 |
| TIES-Merging (w/ DARE) | 11.8 | 95.5 | 85.8 | 9.4 | 86.8 | 88.7 | 83.1 | 63.6 | 65.6 |
| WUDI-Merging | 81.8 | 98.3 | 78.7 | 60.5 | 92.7 | 90.5 | 93.3 | **86.4** | 85.3 |
| **SA–Merging** (Ours) | **85.0** | **98.8** | **82.0** | **70.0** | **93.5** | **91.5** | **94.0** | 82.0 | **87.1** |

**Computational cost comparison.** In Table B4, we provide the comparison of different merging approaches, including TIES-Merging (Yadav et al., 2023), AdaMerging (Yang et al., 2023), WUDI-Merging (Cheng et al., 2025), and our **SA–Merging**, in terms of accuracy, time, and GPU memory usage with ViT-B/32 on the 8-task vision suite. The results show that SA-Merging achieves the best performance of 85.9% with 20s runtime, which is 6× faster than WUDI-Merging. However, SA-Merging keeps the task vectors, the connectivity scores for each expert, and their gradients during the merging process, requiring about 3× more GPU memory than WUDI-Merging. While TIES-Merging is a very lightweight one-shot heuristic, it achieves lower performance than other methods. Our SA-Merging remains substantially more efficient than calibration-based AdaMerging, and is also faster than WUDI-Merging, while delivering the best accuracy.

*Table B3.* Per-task multi-task performance when merging RoBERTa-Large experts on the 8-task GLUE benchmark.

| Method | CoLA | SST-2 | MRPC | STS-B | QQP | QNLI | MNLI | RTE | Avg. |
|---|---|---|---|---|---|---|---|---|---|
| *Non-merging* | | | | | | | | | |
| Pretrained | 0.0 | 51.5 | 40.9 | 20.9 | 36.4 | 56.0 | 37.6 | 62.4 | 38.2 |
| Individual | 100.0 | 100.0 | 100.0 | 100.0 | 100.0 | 100.0 | 100.0 | 100.0 | 100.0 |
| *Data-free model merging* | | | | | | | | | |
| Weight Averaging | 7.4 | 55.1 | 84.2 | 46.3 | 56.7 | 73.8 | 35.8 | 66.7 | 53.3 |
| Task Arithmetic | 7.4 | 86.1 | 86.8 | 78.0 | 90.7 | 77.0 | 73.3 | 67.6 | 70.9 |
| TIES-Merging | 42.7 | 78.1 | 85.2 | 51.7 | 89.9 | 81.9 | 79.7 | 70.0 | 72.4 |
| Task Arithmetic (w/ DARE) | 4.1 | 85.2 | 85.8 | 71.6 | 91.3 | 85.6 | 75.2 | 68.1 | 70.9 |
| TIES-Merging (w/ DARE) | 2.9 | 90.4 | 86.8 | 75.4 | 92.4 | 86.4 | 79.0 | 69.1 | 72.8 |
| WUDI-Merging | 82.2 | 98.7 | 87.3 | 81.4 | 94.6 | **96.6** | 93.4 | **77.1** | 88.8 |
| **SA-Merging** (Ours) | **86.5** | **99.0** | **88.5** | **85.0** | **95.0** | 96.5 | **94.0** | **77.1** | **90.2** |

*Table B4.* Comparisons in computational cost. We report the performance on 8-task vision suite, required time, and GPU memory for TIES-Merging (Yadav et al., 2023), AdaMerging (Yang et al., 2023), WUDI-Merging (Cheng et al., 2025), and our **SA-Merging**.

| Method | Accuracy (%) | Time | GPU Memory (GB) |
|---|---|---|---|
| TIES Merging | 72.4 | 4s | 0 |
| AdaMerging | 81.1 | 127min | 17.1 |
| WUDI-Merging | 85.2 | 1min 54s | 4.0 |
| **SA-Merging** (Ours) | **85.9** | 20s | 14.5 |

