# OpenReview forum: "Saliency-Aware Model Merging"
_ICML.cc/2026/Conference — ICML 2026 regular_

### Official Review · Reviewer_JSiF · 2026-03-13

**Soundness:** 2
**Presentation:** 3
**Significance:** 2
**Originality:** 3
**Overall Recommendation:** 4
**Confidence:** 3

**Summary:**

This paper proposes a sensitivity-aware model merging method that reduces merging interference by pruning parameters based on their importance in cross-layer connections.

**Compliance With Llm Reviewing Policy:**

Affirmed.

**Final Justification:**

The authors have addressed my concerns, and I maintain my previous score.

**Key Questions For Authors:**

There is a large amount of blank space on pages 6 and 7.

**Limitations:**

yes

**Strengths And Weaknesses:**

Strengths:
- This paper proposes a data-free proxy metric to measure the saliency score of parameters in connections.
- The proposed method is applicable to both dense model merging and LoRA model merging.


Weaknesses:
- The proxy metric in Equation (3) is heuristic, lacks a deeper explanation, and cannot guarantee an accurate measurement of parameter importance.
- Equation (4) involves gradient computation; however, the paper does not analyze the memory consumption or the time required for iterative training.
- The experiments in this paper only use relatively small-scale models, such as RoBERTa and Flan-T5-base, and lack validation on LLMs.

---

> ### Author Rebuttal · Authors · 2026-03-31
>
> **(1) Deeper analysis on connectivity**
>
> We thank the reviewer for the thoughtful feedback.
> As we discussed in the reviewer m8Fv's questions in **'Connectivity score as a good proxy'**, we agree that Eq. (3) is a structural surrogate, not an exact model of forward signal propagation.
> Our intention is not to model task-wise activations in inference time, but to provide a data-free, differentiable proxy that captures how strongly parameters participate in end-to-end layerwise paths, inspired by path-norm and SynFlow formulations.
> In addition, the gradient admits a prefix-suffix factorization, as shown in Appendix A.3: the sensitivity of a parameter becomes larger when it connects large upstream and downstream flows.
> Therefore, the score captures whether an update lies on strong end-to-end pathways, rather than only whether its own magnitude is large.
> Furthermore, as we answered in m8Fv's question on **'Ablation study for saliency score formulation'**, the connectivity gradient and the aggregated task vector complement each other, showing the best performance rather than a magnitude-based score.
>
> **(2) Computational cost comparison**
>
> We thank the reviewer for a valuable comment.
> We provide the comparison of different merging approaches in terms of accuracy, time, and GPU memory usage with ViT-B/32.
>
> | Method | Accuracy (%) ↑ | Time ↓ | GPU Memory (GB) ↓ |
> | --- | --- | --- | --- |
> | TIES Merging | 72.4 | 4s | 0 |
> | AdaMerging | 81.1 | 127min | 17.1 |
> | WUDI-Merging | 85.2 | 1min 54s | 4.0 |
> | **SA-Merging (Ours)** | **85.9** | 20s | 14.5 |
>
> The results show that SA-Merging achieves the best performance of 85.9\% with 20s runtime and 14.5GB GPU memory.
> As the reviewer pointed out, SA-Merging requires gradient computation of each expert's connectivity score during the merging process, requiring about $3\times$ more GPU memory than WUDI-Merging.
> However, our SA-Merging remains substantially more efficient than calibration-based AdaMerging, and is also faster than WUDI-Merging which requires 300 steps for iterative optimization, while delivering the best accuracy.
> We will include runtime and memory analysis in the manuscript.
>
> **(3) Large-scale LLM experiment**
>
> We appreciate your valuable comment.
> We additionally evaluate SA-Merging on Qwen-14B LoRA experts across four tasks, as shown in the table below.
>
> | Method | MMLU | TruthfulQA | BBQ | CNN | Avg. |
> | --- | --- | --- | --- | --- | --- |
> | Individual | 68.35 | 53.34 | 93.53 | 19.46 | 58.67 |
> | Task Arithmetic | 67.56 | 52.33 | 78.38 | **20.54** | 54.70 |
> | TIES Merging | 69.38 | 52.03 | 81.06 | 15.91 | 54.62 |
> | WUDI-Merging | 69.17 | **55.71** | 80.56 | 17.33 | 55.69 |
> | **SA-Merging (Ours)** | **69.87** | 55.60 | **81.35** | 18.48 | **56.33** |
>
> The result shows that SA-Merging consistently outperforms the baselines, supporting that the proposed rank-wise saliency formulation is not limited to Flan-T5-base, but generalizes across model sizes as well.
> We will add this result to the manuscript.
>
> If there are further experimental results or analyses that reviewers would find helpful during the discussion period, we would welcome the opportunity to provide them to the best of our ability.
>
> **(4) Formatting issue**
>
> We thank the reviewer for pointing out the format issue. We will adjust spacing in the revised version.

---

> > ### Author Rebuttal · Reviewer_JSiF · 2026-04-02
> >
> > Thank you for your response. Since my score was already positive before, I will maintain this score after reading your reply.

---

> > > ### Author Response · Authors · 2026-04-05
> > >
> > > We sincerely thank the reviewer for the continued support of our work. Your detailed and constructive feedback throughout the review process has meaningfully improved the paper.

---

### Official Review · Reviewer_m8Fv · 2026-03-13

**Soundness:** 3
**Presentation:** 2
**Significance:** 3
**Originality:** 3
**Overall Recommendation:** 4
**Confidence:** 3

**Summary:**

This paper introduces SA-Merging, a data-free model merging method that aims to consolidate multiple task-specific models into a single one. The core idea is to estimate the saliency of each parameter by defining a differentiable inter-layer interaction function. The gradients of this function are used to derive a saliency score, which identifies parameters crucial for maintaining end-to-end information flow. Based on this score, the method iteratively prunes non-informative parameters before merging. The authors also extend this approach to handle LoRA adapters by proposing a rank-wise saliency decomposition. The method is evaluated on various vision and language tasks, showing improved performance over existing data-free merging techniques.

**Compliance With Llm Reviewing Policy:**

Affirmed.

**Key Questions For Authors:**

Please refer to the weaknesses

**Limitations:**

Yes

**Strengths And Weaknesses:**

**Strengths**
- The core idea of using a data-free, differentiable proxy for inter-layer connectivity to guide model merging is interesting. It moves beyond simple parameter-level heuristics and attempts to incorporate structural information, which is a sensible direction for improving model merging.
- The proposed method is evaluated across a reasonable range of tasks and model architectures, including vision (ViT) and language (RoBERTa, T5) models. The inclusion of both full-parameter and LoRA-based merging demonstrates the versatility of the approach. The experimental results are strong. This suggests the saliency signal is effective.
- The extension to LoRA merging is a valuable contribution. Preserving the low-rank structure during merging is a non-trivial problem, and the proposed rank-wise saliency decomposition (Section 3.4) is a thoughtful approach to address this, avoiding the need to materialize the full weight matrices.

**Weaknesses**
- The central component of the method, the connectivity score $\mathcal{R}_n$ in Equation (3), is not well-justified, and its definition raises several questions.

  - The function $\mathcal{R}{n}(\theta{0},\tau_{n}):=\mathbb{1}^{\top}\left(\prod_{l=1}^{L}\left|\theta_{0}^{l}+\tau_{n}^{l}\right|\right)\mathbb{1}$ is presented as a proxy for "total signal transmission capacity". However, the product of weight matrices does not directly correspond to signal flow in a standard feed-forward network, which involves matrix-vector products with activations. This formulation seems more inspired by the path-norm ideas, but the connection is not rigorously established. Why is this specific product of absolute-valued matrices a good proxy for functional importance?
  - The use of all-ones vectors $\mathbb{1}$ to sum up the final matrix entries is heuristic. What is the intuition behind this? It seems to treat all output dimensions as equally important, which is unlikely to be true.

- The saliency score in Equation (4), $\mathcal{S}{n}:=\frac{\partial\mathcal{R}{n}}{\partial\tau_{n}}\odot\left|\sum_{i=1}^{N}\tau_{i}\right|$, has conceptual problems.
  - The paper claims the modulation with $|\sum \tau_i|$ performs "implicit sign election". However, it uses the absolute value of the sum. This means a parameter update $\tau_{n,j}$ that strongly disagrees with the consensus direction (e.g., $\tau_{n,j} = -10$ while $\sum \tau_{i,j} = 10$) would have its gradient's magnitude amplified, not suppressed. The term $\frac{\partial\mathcal{R}{n}}{\partial\tau{n}}$ can be positive or negative. If it has the same sign as $\tau_{n,j}$, and $\tau_{n,j}$ has the opposite sign of $\sum \tau_{i,j}$, the final saliency might not be small. A more direct sign agreement mechanism, like $\text{sign}(\tau_{n,j}) \cdot \text{sign}(\sum \tau_{i,j})$, as used in TIES-merging, seems more appropriate for "sign election". The authors should clarify this mechanism or correct the formulation.
  - In Algorithm 1, line 6, the saliency is computed as $\mathcal{S}{n}\leftarrow\frac{\partial\mathcal{R}{n}}{\partial\tau_{n}}\odot\tau_{t}^{\star}$, where $\tau_{t}^{\star} = \sum_i \tau_i$. This differs from Equation (4), which uses the absolute value $|\sum \tau_i|$. This is a critical inconsistency. If the absolute value is not used, then a parameter with a negative gradient and a negative aggregate direction would get a large positive saliency, which might be intended, but this contradicts the description of "implicit sign election" and the use of absolute value in Equation (4). This needs to be clarified and justified.

- The analysis in Section 4.4 is superficial and does not provide deep insights into why the method works.

  - The core claim is that SA-Merging is superior because it considers inter-layer connectivity. However, there is no ablation to disentangle the effects of the two components of the saliency score: the connectivity gradient ($\partial\mathcal{R}_n/\partial\tau_n$) and the interference term ($|\sum \tau_i|$). How does a variant using only the connectivity gradient perform? How about a variant that combines a standard magnitude-based score with the interference term? This would help isolate the true source of the performance gain.
  - Figure 3(c) shows that SA-Merging reduces sign conflicts compared to TIES-Merging. This is an interesting result, but the y-axis is "Frac. of Params with Conflict". How is "conflict" defined here? Is it simply when $\text{sign}(\tau_{n,j}) \neq \text{sign}(\sum_{i \neq n} \tau_{i,j})$? Given the potential issues with the sign agreement mechanism (Weakness #2), this claim requires more rigorous support and a clearer definition.
  - The analysis of pruning ratio and iterations in Figure 3(a) and 3(b) is standard but doesn't offer much insight. For instance, why does performance on DTD drop so sharply for $p=0.2$ in Figure 3(a)? The analysis lacks depth.

---

> ### Author Rebuttal · Authors · 2026-03-31
>
> **Connectivity score as a good proxy**
>
> We agree that Eq. (3) is a structural surrogate, not an exact model of forward signal propagation. Its purpose is not to model task-specific activations at inference time, but to provide a data-free and differentiable proxy for how strongly parameters participate in end-to-end layerwise paths, inspired by path-norm and SynFlow. As shown in Appendix A.3, the gradient admits a prefix-suffix factorization, so a parameter receives a larger score when it connects strong upstream and downstream flows. Thus, the score reflects whether an update lies on important end-to-end pathways, not merely whether its magnitude is large.
>
> The use of the all-ones vectors is a uniform aggregation choice, not a claim that all output coordinates are semantically equally important. Their role is to yield the matrix-valued path-strength quantity to a scalar objective in a \textit{fully data-free way}. We will clarify this design choice in the manuscript.
>
> **Equation correction**
>
> We apologize for the careless writing and appreciate for pointing out this. The saliency score formulation modulates the connectivity gradient with the signed aggregate direction so that coordinates conflicting with the ensemble direction are down weighted, exactly as described in the main paper. We will revise Eq. (4) as $\mathcal{S}_n := \frac{\partial \mathcal{R}_n}{\partial \tau_n} \odot \sum_i  \tau_i$ so that the equation, algorithm, and text are fully consistent.
>
> **Implicit sign election**
>
> We thank the reviewer for this important discussion. Since the saliency score is computed by taking the partial derivative of the connectivity score $\mathcal{R}_n$ with respect to $\tau_n$, the resulting $\mathcal{S}_n$ naturally includes the sign of $\tau_n$. Therefore, the interaction between $\operatorname{sign}(\tau_n)$ and $\sum_i \tau_i$ can be understood as an implicit sign election mechanism.
> For the simplest example, consider a network with three linear layers parameterized by $\mathbf{W}_n^1$, $\mathbf{W}_n^2$, and $\mathbf{W}_n^3$, where $\mathbf{W}_n^{(l)} = \mathbf{W}_0^{(l)} + \tau_n^{(l)}.$
> Then, the saliency score of the $(j,k)$-th component in $\mathbf{W}_n^1$ is
>
> $\mathcal{S}_{n,jk}^1$
>
> $=\operatorname{sign}(\tau_{n,jk}^1)\cdot \left[\sum_i \tau_i^1\right]_{j,k} \cdot \left[\left((|\mathbf{W}_n^3||\mathbf{W}_n^2|)^\top \mathbf{1}\right)\right]_j.$
>
> This formulation indicates the following:
>
> - When first two terms have the same sign, $\tau_{n,jk}$ is aligned with the dominant direction of the merged model, yielding a positive saliency score.
> - When they have opposite signs, $\tau_{n,jk}$ disagrees with the direction of the merged model, yielding a negative saliency score.
> - When multiple models cancel each other out in $\[\sum_i \tau_i^1\]_{j,k}$, the saliency score naturally becomes small, approaching zero.
>
> We will revise the manuscript to make this implicit sign election clear.
>
> **Ablation study for saliency score formulation**
>
> As the connectivity gradient and the aggregated task vector complement each other, as answered in the above (4) question, removing either part weakens the formulation in a different way.
>
> |  | Saliency score | ViT-B/32 Acc. |
> | --- | --- | --- |
> | (a) | $ \| \tau_n \| \odot \sum_i \tau_i $ | 72.5 |
> | (b) | $\frac{\partial \mathcal{R}_n}{\partial \tau_n}$ | 71.7 |
> | (c) | $\frac{\partial \mathcal{R}_n}{\partial \tau_n} \odot \sum_i \tau_i$ | 85.9 |
>
> Using only the connectivity gradient (row (b)) captures structural sensitivity, but ignores agreement with the consensus merge direction, which can retain conflicting coordinates. In contrast, row (a) reflects direction but misses the inter-layer dependency captured by the gradient, overemphasizing locally dominant coordinates. Accordingly, both ablations underperform the full formulation. We will include this ablation and discussion in the manuscript to clarify the role of each component.
>
> **Sign conflicts**
>
> Following TIES Merging, a sign conflict is defined at coordinate $j$ if there exist two task vectors $n \neq m$ such that their final updates have opposite signs. Therefore, Fig. 3(c) shows that the parameter selection induced by our method leaves fewer mixed-sign coordinates than magnitude-based trimming, which supports that SA-Merging identifies merge-compatible parameters more effectively than magnitude-based trimming in TIES Merging.
>
> **Effect of pruning ratio and iterations**
>
> SVHN, GTSRB, MNIST, and DTD are more sensitive than the other datasets, likely because they rely on specialized, less redundant cues: fine shape and stroke patterns in SVHN, GTSRB, and MNIST, and texture statistics rather than object-level semantics in DTD. These tasks benefit more from progressive refinement, since one-shot coarse pruning can remove task-specific coordinates too early. As $T$ increases, the saliency score is recomputed after each step, allowing the mask to adapt more gradually. We will revise Sec. 4.4.1 to clarify this point.

---

> > ### Author Rebuttal · Reviewer_m8Fv · 2026-04-04
> >
> > Thank you for the constructive rebuttal. The responses mostly resolve my concerns. At this time, I will keep my current score.

---

> > > ### Author Response · Authors · 2026-04-05
> > >
> > > We sincerely thank the reviewer for the continued support of our work. Your detailed and constructive feedback throughout the review process has meaningfully improved the paper.

---

### Official Review · Reviewer_LChx · 2026-03-13

**Soundness:** 3
**Presentation:** 3
**Significance:** 3
**Originality:** 3
**Overall Recommendation:** 4
**Confidence:** 3

**Summary:**

This paper proposes Saliency-Aware Model Merging (SA-Merging), a data-free framework for model merging. Previous model merging methods ignore the dependencies between model layers. To address this, SA-merging uses a differentiable connection function to estimate the significance of parameters as weights. Additionally, this paper designs a rank-wise saliency score to extend SA-Merging to LoRA adapters. Experiments show that SA-merging achieves significant performance improvements on both visual and language tasks.

**Compliance With Llm Reviewing Policy:**

Affirmed.

**Final Justification:**

The rebuttal removes most of my concerns and I maintain the original score.

**Key Questions For Authors:**

Did the authors study the impact of different pre-training models on model merging? For instance, could we replace the CLIP-based model with the DINO series or the MAE series? Is SA-merging still effective for other pre-training models?

**Limitations:**

Yes

**Strengths And Weaknesses:**

Strengths
- The problem addressed in this article is highly meaningful. SA-merging overcomes the previous methods' neglect of the inter-layer relationships.
- SA-merging can be extended to LoRA models through the rank-wise saliency.
- Experimental evaluation is conducted across both vision models and language models.

Weaknesses
- This paper lacks a quantitative comparison of the running time/computational cost with existing methods. Although the authors mentioned the issue of cost in the Limitations section, no actual data are provided to support this.
- The connectivity function is based on the product of consecutive layers, assuming the chain-like reasoning of the model. It remains unclear whether this formulation can accommodate architectures with dense skip connections, such as ResNet or DenseNet.
- LORA has also been widely used in visual models, particularly ViT models. This paper only covers the extension experiments of LoRA for language models.

---

> ### Author Rebuttal · Authors · 2026-03-31
>
> **(1) Computational cost comparison**
>
> We thank the reviewer for a valuable comment. We provide the comparison of different merging approaches in terms of accuracy, time, and GPU memory usage with ViT-B/32.
>
> | Method                | Accuracy (%) ↑ |   Time ↓ | GPU Memory (GB) ↓ |
> | --------------------- | -------------: | -------: | ----------------: |
> | TIES Merging          |           72.4 |       4s |                 0 |
> | AdaMerging            |           81.1 |   127min |              17.1 |
> | WUDI-Merging          |           85.2 | 1min 54s |               4.0 |
> | **SA-Merging (Ours)** |       **85.9** |      20s |              14.5 |
>
> The results show that SA-Merging achieves the best performance of 85.9\% with 20s runtime, which is $6\times$ faster than WUDI-Merging. However, SA-Merging keeps the task vectors, the connectivity scores for each expert, and their gradients during the merging process, requiring about $3\times$ more GPU memory than WUDI-Merging. While TIES-Merging is a very lightweight one-shot heuristic, it achieves lower performance than other methods. Our SA-Merging remains substantially more efficient than calibration-based AdaMerging, and is also faster than WUDI-Merging, while delivering the best accuracy. We will include runtime and memory analysis in the manuscript.
>
> **(2) Applicability to architectures with skip connections**
>
> We thank the reviewer for this important point. While we use a product of consecutive blocks in Eq. (3) for expository simplicity, this formulation is not restricted to chain-structured networks.
> For example, given a fully-connected residual block and input $x$, the intermediate hidden states and the output $y$ can be represented as
>
> $$
> \mathbf{h}_1 = \mathbf{W}^1 \mathbf{x}, \quad
> \mathbf{h}_2 = \mathbf{W}^2 \mathbf{h}_1, \quad
> \mathbf{y} = \mathbf{W}^3 \mathbf{h}_2 + \mathbf{h}_1.
> $$
>
> The connectivity score $\mathcal{R}_n$ is then
> $$
> \mathcal{R}_n := \mathbf{1}^\top (|\mathbf{W}^3||\mathbf{W}^2| + \mathbf{I}) |\mathbf{W}^1| \mathbf{1}.
> $$
>
> The saliency score for the first layer can be derived as follows.
>
> $$\mathcal{S}_n^1=
> \operatorname{sign}(\tau_n^1) \odot \tau_t^{1*} \odot
> \left[
> \left(
> (|\mathbf{W}_n^3||\mathbf{W}_n^2| + \mathbf{I})^\top \mathbf{1}
> \right)
> \mathbf{1}^\top
> \right].
> $$
>
> Therefore, the saliency scores for the second and third layers can also be derived in the same way.
>
> **(3) LoRA experiments**
>
> We agree that extending LoRA merging experiments to vision models would be valuable. However, existing model merging benchmarks and baselines for LoRA are more established in the language model setting, whereas comparable benchmarks for LoRA-tuned ViT models are currently limited. Therefore, for a fair comparison, we evaluated the LoRA extension on Flan-T5-base GLUE experts, following standard data-free model merging benchmarks. To further support generality, we additionally evaluate SA-Merging on Qwen-14B LoRA experts across four tasks, as shown in the table below.
>
> | Method | MMLU | TruthfulQA | BBQ | CNN | Avg. |
> | --- | --- | --- | --- | --- | --- |
> | Individual | 68.35 | 53.34 | 93.53 | 19.46 | 58.67 |
> | Task Arithmetic | 67.56 | 52.33 | 78.38 | **20.54** | 54.70 |
> | TIES Merging | 69.38 | 52.03 | 81.06 | 15.91 | 54.62 |
> | WUDI-Merging | 69.17 | **55.71** | 80.56 | 17.33 | 55.69 |
> | **SA-Merging (Ours)** | **69.87** | 55.60 | **81.35** | 18.48 | **56.33** |
>
> The result shows that SA-Merging consistently outperforms the baselines in this larger-scale PEFT setting, supporting that the proposed rank-wise saliency formulation is not limited to Flan-T5-base, but generalizes across model sizes as well. Combined with the strong results on ViT full-parameter merging and language LoRA merging, the evidence supports the broader claim that SA-Merging is not modality-specific. We will add this result to the manuscript, and include LoRA merging for vision models as future work.
>
> **(4) Different pretraining objectives**
>
> Given that the key requirement for SA-Merging is that fine-tuned experts share the same pre-trained base checkpoint (e.g., ViT architecture), we expect SA-Merging to generalize to other pre-training objectives such as CLIP, DINO, or MAE. In our formulation, the merging procedure is defined entirely on base parameter $\theta_0$ and task vector $\{ \tau_n \}$, i.e., tied more strongly to the shared-base task-vector setting than to CLIP itself. For this reason, the method is, in principle, compatible with other pre-training families as well.

---

> > ### Author Rebuttal · Reviewer_LChx · 2026-04-02
> >
> > Thanks the authors for adding the experiments on computational cost and multimodal models in the rebuttal. The authors' rebuttal has largely addressed my concerns. Although experiments on visual LoRA models and other ViT model (eg. DINO, MAE) are still missing, these are not so critical. Therefore, I maintain my previous score (weak accept).

---

> > > ### Author Response · Authors · 2026-04-05
> > >
> > > We sincerely thank the reviewer for the continued support of our work. Your detailed and constructive feedback throughout the review process has meaningfully improved the paper.

---

### Official Review · Reviewer_8YsC · 2026-03-15

**Soundness:** 3
**Presentation:** 2
**Significance:** 2
**Originality:** 2
**Overall Recommendation:** 4
**Confidence:** 3

**Summary:**

This paper introduces Saliency-Aware Model Merging (SA-Merging), a data-free algorithm for model merging. The core motivation behind the design of SA-Merging is that existing techniques implicitly assume that the saliency of a parameter can be deduced by its value only, thereby overlooking its structural importance in the end-to-end information flow of the network.

SA-Merging computes the importance of each parameter in each expert using the gradient of the end-to-end score that sums the $\ell1$-path norm passing through such a parameter, modulated by the arithmetically merged model. This procedure is repeated iteratively, up to a predefined number of iterations, and at each iteration a fixed fraction of weights is removed (i.e., masked) from each expert. The resulting sparse task vectors and then combined through task arithmetic to yield the final model.

Experiments are conducted on CLIP-trained ViTs, text encoders (RoBERTa-base and -large), and text decoders (Flan-t5), showing that SA-Merging improves over existing methods.

**Compliance With Llm Reviewing Policy:**

Affirmed.

**Final Justification:**

The discussion clarified the two major aspects leading to my initial skepticism: missing credits for existing work and unspecified hyperparameter tuning.
- For the former, the authors acknowledged existing work and provided detailed, ad-hoc edits for specific passages in the manuscript to dispel the initial unintended impression.
- For the latter, the authors clarified that the "data-free" property held during their hyperparameter search as well.

I am left with no major unresolved points, hence I consider it fair and appropriate to improve my recommendation.

**Key Questions For Authors:**

I'd appreciate a discussion about the following items.
- Are the authors willing to properly attribute merits to SynFlow whenever their work uses related concepts and/or algorithms? What are the authors' thoughts about this point, in general?
- Could the authors clarify their hyperparameter selection protocol? On a broader note, I believe data-free methods generally suffer from the implicit limitation of actually needing validation data despite claiming to be data-free, which seems to be the case for this work as well. What are the authors' thoughts regarding this aspect?

I am willing to increase my score if the above points are cleared.

**Limitations:**

Yes, the paper discussed limitations and negative societal impacts properly, which I appreciate.

**Strengths And Weaknesses:**

**Strengths.**
- The work proposes a data-free algorithm for model merging, following the line of works that do not rely on calibration samples to increase the practicality of merging techniques;
- The paper proposes an extension of SA-Merging for LoRA adapters, which are typically used to obtain experts from base models;
- The experimental section is comprehensive and provides a broad spectrum of both vision and language tasks;

**Weaknesses.**
- The biggest weakness of this work, in my opinion, is that the proposed method is essentially an exact application of SynFlow (a well-known pruning algorithm [a]) to a merged model. I'd like to avoid potential comments on "novelty", as they are often subjective and therefore don't fit well in scientific reviews in my opinion, and I further believe that combining different fields might be valuable in general. What puzzles me is the fact that this paper seems to ignore the existence of such a work, and claims theoretical or methodological contributions while they might be just direct applications of existing ideas. For example:
    - The score $\mathcal{R}_n$ computed for each expert is taken exactly from SynFlow, yet the paper claims (lines 138-139, right column) *"[...] we introduce a connectivity-based saliency score"*;
    - Similarly, the iterative procedure is also taken from SynFlow, yet the paper claims (lines 195-196, left column): *"we propose an iterative scheme to progressively refine [...]"*.
- The claim *"Our approach bypasses the need for [...] or even unlabeled calibration data to determine hyperparameters or importance weights."* (lines 221-224, right column) appears to be an overclaim. By looking at Figures 3(a) and 3(b), one could see that the proposed SA-Merging would lag far behind other merging strategies if hyperparameters had not been tuned carefully, and that SA-Merging itself is quite sensitive to such hyperparameter choices.
- Connecting to the previous point, it is quite unclear how this work has tuned hyperparameters for their method. By looking at Figures 3(a) and 3(b), it appears that the best values have been taken by maximizing performance on the exact test sets that are then used for comparison.

**Minor Weakness.**
- I believe Equation (4) (saliency score $\mathcal{S}_n$, which also appears identical to SynFlow), might be wrong, i.e., it modulates the estimated importance with the magnitude of the merged model, while the text and Algorithm 1 suggest the sign is taken into account (i.e., no magnitude).



**References.**\
[a] Tanaka, Hidenori, et al. "Pruning neural networks without any data by iteratively conserving synaptic flow." Advances in neural information processing systems 33 (2020).

---

> ### Author Rebuttal · Authors · 2026-03-31
>
> We sincerely thank the reviewer for the careful reading and insightful comments. We especially appreciate the constructive feedback regarding the relation to SynFlow and the clarification of hyperparameter selection. We address these points below.
>
> **(1) Relation to SynFlow and novelty clarification**
>
> We fully agree that our connectivity-based saliency formulation is closely related to SynFlow. We first clarify that the current draft already cites Tanaka et al. (2020) and explicitly states that our method is inspired by structural pruning (L.91-95). However, we agree with the reviewer that this connection was not sufficiently emphasized, and that our current presentation may give the impression of introducing the underlying saliency formulation as entirely new. We will revise the paper to more clearly and prominently position SynFlow as a key foundation of our approach.
>
> We would like to clarify that the primary contribution of this work is the extension of SynFlow for data-free model merging, which introduces several unique challenges beyond standard single model pruning:
> - **Cross-task interference**: unlike pruning, model merging requires resolving conflicting updates across multiple experts. In our setting, the connectivity gradient is defined by a partial derivative with respect to each task vector (instead of the whole fine-tuned weight) and further combined with the current aggregated merge direction to produce a interference-aware saliency that reflects both structural (inter-layer) importance and inter-expert agreement, which is different from applying SynFlow to a single standalone network.
> - **Compatibility**: in our work, saliency must be computed on task vectors relative to a shared base, not on a single model.
> - **Iterative masking**: while iterative pruning exists in SynFlow (we will explicitly make that attribution), our formulation operates on evolving aggregated direction, which changes the saliency landscape across iterations.
> - **LoRA extension**: we introduce the rank-preserving LoRA merging, where saliency is defined over rank-1 components and masking is performed without destroying the low-rank parameterization. It is not addressed in SynFlow.
>
> Therefore, we will revise the manuscript to avoid overstating novelty and more precisely position our work as follows: (1) formulating the connectivity-based saliency of SynFlow to data-free model merging; (2) introducing a merge-aware saliency modulation for interference reduction; and (3) reformulating the same principle for LoRA rank-wise model merging.
>
> **(2) Hyperparameter selection**
>
> We apologize for the lack of clarity regarding hyperparameter selection. The hyperparameters are fixed a priori by a data-free rule, and not tuned per benchmark. Concretely, our protocol is as follows. We first set the number of iterations to $T=10$ as a fixed practical default. As shown in Figure 3(b), we observe a consistent trend that performance improves as $T$ increases, across all tasks, indicating that iterative refinement is beneficial and not sensitive to task-specific tuning.
> Furthermore, we observe that this trend holds consistently across different pruning ratios $p$. Following prior sparsification and localized-merging literature such as TIES-merging and Localize-and-stitch, we then set a target number of parameters to keep after $T$ iterations to 10\% of the total number of each task vector's parameters. Therefore, the iteration pruning ratio is set to $p=0.2$ to satisfy $(1-p)^T \approx 0.1$.
>
> In addition, while Figures 3 (a) and (b) show that the fixed default (i.e., $p=0.2$, $T=10$) lies in the strong-performance region and that performance stabilizes near 10 iterations, these sweeps were not used to select hyperparameters on any validation sets. In practice, we found the maximum performance of 87.0\% of ViT-B/32 with $p=0.06$ and $T=20$ using a small validation subset. We will include a clear explanation of the hyperparameter selection protocol in the manuscript.
>
> More broadly, we agree with the reviewer that the term "data-free" can sometimes be ambiguous in two points: whether the merge-time algorithm uses data, and whether a practitioner may still use validation data for model selection. Our claim includes both points in that SA-Merging computes saliency entirely from parameters, and the hyperparameters ($p$ and $T$) are set without using validation performance.
>
> **(3) Equation (4) and consistency with Algorithm 1**
>
> We apologize for the careless writing and appreciate for pointing out this. We will revise Eq. (4) as $\mathcal{S}_n := \frac{\partial \mathcal{R}_n}{\partial \tau_n} \odot \sum_i \tau_i$, which modulates the gradient of connectivity with the signed aggregate direction so that coordinates conflicting with the ensemble direction are down weighted, exactly as described in the main paper.

---

> > ### Author Rebuttal · Reviewer_8YsC · 2026-04-03
> >
> > Dear Authors,
> >
> > Thank you for sharing your hyper-parameter selection protocol. I believe it is valid, and I would recommend enriching the manuscript accordingly to let readers know how to reproduce your work in all of these not-so-minor details.
> >
> > Regarding the analogies and differences with SynFlow: I am glad to see you recognize the merits of such work, and that you honestly admit that the current draft might give the impression of overstating novelty in some passages. Would you mind posting a response with the ad-hoc modifications that you plan to incorporate in the manuscript?
> >
> > Best,\
> > Reviewer `8YsC`

---

> > > ### Author Response · Authors · 2026-04-05
> > >
> > > We appreciate the constructive follow-up. Below, we provide the sentence- and paragraph-level changes we plan to incorporate.
> > >
> > > We will revise
> > >
> > > 1. L018-031 (left column):
> > > > In this work, we build upon connectivity-based saliency formulations from structural pruning (e.g., SynFlow) and extend them to the data-free model merging setting. We define a saliency score over task vectors relative to a shared base model, and further introduce merge-aware modulation that incorporates agreement across experts to mitigate interference.
> > > Based on this formulation, we develop an iterative saliency-aware merging procedure that progressively removes non-informative updates while preserving end-to-end connectivity.
> > >
> > > 2. L089-095 (left column):
> > > > In this work, we introduce an iterative saliency-aware model merging (SA-Merging) that incorporates the connectivity criterion for single network pruning in SynFlow (Tanaka et al., 2020), which has demonstrated the inter-layer interactions of parameters can be measured along a path from an input to an output node, into multiple pre-trained model merging.
> > >
> > > 3. L096-099 (left column) :
> > > > Following prior work on connectivity-based pruning (Tanaka et al., 2020), we define a connectivity score as a structural proxy for task-wise end-to-end influence.
> > >
> > > 4. L138-139 (right column):
> > > > Motivated by SynFlow's connectivity score, where saliency is computed for a single model, we reformulate the connectivity-based saliency score with respect to task vectors and further modulate it using the aggregated merge direction to account for cross-expert agreement.
> > >
> > > 5. L195-196 (left column):
> > > > we follow the iterative pruning scheme in (Tanaka et al., 2020) to progressively refine the merged parameter mask.
> > >
> > > 6. L217-219 (left column):
> > > > We first set the number of iterations $T$ to 10 as a fixed practical default. As shown in Figure 3(b), we observe a consistent trend that performance improves as $T$ increases across all tasks. Furthermore, we observe that this trend holds consistently across different pruning ratios. Following prior sparsification and localized-merging literature such as TIES-merging and Localize-and-stitch, we then set a target number of parameters to keep after iterations to 10\% of the total number of each task vector's parameters. Therefore, the iteration pruning ratio is set to $p=0.2$ to satisfy $(1-p)^T \approx 0.1$.
> > >
> > > 7. L397-401 (right column):
> > > > We utilized the connectivity-aware saliency basis of the task vector that couples consecutive layers and yields parameter rankings that reflect end-to-end influence, and iterative saliency pruning that constructs a refined task-wise parameter mask for composing task vectors.
> > >
> > > We would welcome the reviewer's suggestions for revising the paper.
> > >
> > > **Thank you for the discussion and feedback.**
> > >
> > > Your feedback was very helpful in improving both the technical positioning and the clarity of the manuscript.
> > >
> > > Following your comments, we clarified the hyperparameter selection protocol in a reproducible manner and provided concrete manuscript-level revisions to more explicitly attribute the connectivity criterion and iterative pruning scheme to SynFlow, while more carefully positioning our contribution as an extension to data-free model merging and LoRA merging.
> > >
> > > If you believe that these clarifications and planned revisions have sufficiently addressed your concerns, we would be grateful if you would consider reflecting that in your final assessment.
> > >
> > > Thank you again for your careful review and valuable feedback.

---

### Decision · Program_Chairs · 2026-04-30

**Decision:**

Accept (regular)

**Comment:**

This paper proposes a data-free model merging method based on connectivity-aware saliency, and extends the approach to LoRA via rank-wise saliency. Reviewers agreed that the problem is important and that the paper presents strong empirical results across both vision and language tasks, including a useful extension to LoRA merging.

The rebuttal well addressed the major concerns on the paper’s positioning and technical clarity, and all reviewers remained positive after rebuttal.

Overall, while the paper should more clearly acknowledge its connection to prior pruning work, it makes a solid contribution to data-free model merging and LoRA merging, supported by strong experiments.  The AC recommends acceptance.